# Experiential Avoidance Mediates the Relationship between Prayer Type and Mental Health before and through the COVID-19 Pandemic

Gabriel B. Lowe [1,*], David C. Wang [2] 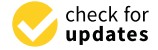 and Eu Gene Chin [3] 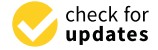

1   Independent Researcher, Encinitas, CA 92024, USA
2   Fuller Theological Seminary, Pasadena, CA 91182, USA; david.wang@biola.edu
3   Rosemead School of Psychology, Biola University, La Mirada, CA 90639, USA; eu.gene.chin@biola.edu
*   Correspondence: g.lowe3290@gmail.com

**Abstract:** The practice of prayer has been shown to predict various mental health outcomes, with different types of prayer accounting for different outcomes. Considering the numerous stressors facing seminary students, which have only intensified throughout the COVID-19 pandemic, prayer may be a common coping strategy for students who study theology, spiritual growth, and leadership. The present study investigates the role that different types of prayer may have in reducing anxiety, depression, and work burnout among seminary students. Experiential avoidance is proposed as a mediator such that specific types of prayer contribute to greater spiritual and characterological formation through staying engaged in the midst of struggle. Longitudinal data was collected from 564 graduate seminary students from 17 institutions accredited by the Association of Theological Schools. Based on previous research, we hypothesized that experiential avoidance would mediate the relationship between colloquial, liturgical, meditative, and petitionary prayer types and the negative mental health outcomes of anxiety, depression, and work burnout. Results confirmed significant negative relationships between colloquial, liturgical, and meditative prayer types and all three mental health indicators, fully mediated by experiential avoidance. Petitionary prayer was not significantly related to lower levels of mental health. These results indicate that engaging in certain prayer practices may be a protective factor by facilitating experiential engagement.

**Keywords:** prayer type; experiential avoidance; seminarians; anxiety; depression; burnout; COVID-19

## 1. Experiential Avoidance as a Mediator between Prayer Type and Mental Health in a Seminary Sample Introduction

Clergy operate at the front lines and play vital roles in fostering healthy communities, especially among minority groups (Wang et al. 2003) and elderly community members (Weaver et al. 2003). On any given workday, clergy members may serve as ritualists, pastors, preachers, teachers, organizers, and/or administrators (Milstein et al. (2005)). Consequently, they face many occupational-type hazards due to the complex and public-facing nature of their roles, including presumptive expectations, personal criticisms, ambiguous boundaries (Lee and Iverson-Gilbert 2003), loneliness (Terry and Cunningham 2020) and stress impinged upon their own family connections (Darling et al. 2004). Navigating these powerful interactive systems (e.g., family pressures, peer expectations, congregant needs) can be especially challenging and may downregulate the well-being of clergy and congregant members (Grosch and Olsen 2000), resulting in shame and poor mental health (Blea et al. 2021).

Prior to being clergy, many individuals seek training from seminary institutions to prepare them for the roles, responsibilities, and challenges of this vocation. In these settings, students are encouraged to grow in their theological understanding, professional skills, and spiritual development. Spiritual growth among seminarians is challenging, with

many citing the rigor of their academic demands and juggling multiple responsibilities as impediments toward formation of their spiritual lives (Greenman and Siew 2001). In service of spiritual formation, clergy are often expected to practice and develop prayer on their own. Broadly defined as a spiritual or religious act of communication between humans and the divine, Poloma and Pendleton (1989) describe four types of prayers (i.e., petitionary, colloquial, ritualistic, and meditative). Several studies show that prayer predicts enhanced mental health through various mechanisms (e.g., feeling like God is in control; Jeppsen et al. 2015), but none have tested experiential avoidance as a potential causal mechanism.

Experiential avoidance, defined as value-incongruent attempts to escape from or control uncomfortable or undesirable internal states (e.g., thoughts, feelings), is garnering increasing amounts of empirical support as a significant contributor to psychopathology and other negative outcomes, including among samples of religious leaders (Jankowski et al. 2022b). In a systematic review, Chawla and Ostafin (2007) found that experiential avoidance was associated with trauma symptoms, substance use relapse, maladaptive coping, poor self-regulation, and overall mental illness severity. This study contributes to the literature in a few ways. First, longitudinal data obtained from seminarians across two years of data collection will establish the temporal precedence of experiential avoidance when examining indirect effects of prayer on mental health outcomes. Second, as opposed to broad operationalizations of mental health in previous studies (Black et al. 2015; Jeppsen et al. 2015), this study tracks more specific indices of mental health (i.e., anxiety, depression, and burnout symptoms) to provide greater specificity in tested models. Results of the study are discussed in relation to empirical and theological literature, with suggested future research directions to broaden the scope and deepen our understanding of prayer and well-being among clergy members.

*Clergy Roles and Responsibilities*

Clergy often fill many roles within their communities, both explicitly and implicitly. Milstein and his colleagues (2005) operationally defined six roles clergy frequently assume to varying degrees. As a *ritualist*, they guide ritual practices (e.g., sacraments, ordinances, etc.), preside over rites of passage (e.g., confirmation, weddings, funerals, etc.), and lead worship services. The role of *pastor* refers to the relational duties of caring for the needs of the congregation at an individual level, whereas the role of *preacher* refers to the inspirational and guiding duties of caring for the needs of the congregation at the group level. One of the more apparent roles is that of *teacher*, which refers to explicit activities such as preaching, teaching, and educating. Behind the scenes, clergy also operate as *organizers* who coordinate communal activities within the organization as well as engage with their larger community, and as *administrators* who oversee the logistics of running their religious establishment.

In addition to the wide skill set expected of clergy, they face specific stressors within these varied roles, especially given the public-facing nature of their position. Some of the occupational hazards identified by previous researchers of this population include unsolicited criticism, boundary ambiguity, presumptive expectations (Frenk et al. 2013; Lee and Iverson-Gilbert 2003), loneliness and lack of social support (Darling et al. 2004; Frenk et al. 2013; Terry and Cunningham 2020), and burnout (Grosch and Olsen 2000). These stressors extend to the families of clergy as well. Darling and her colleagues (2004) found that clergy spouses showed similar risk levels for compassion fatigue, and burnout and experienced higher levels of psychological and physiological stress compared to those found in clergy. High workload, deep emotional demands, and low social support are strong predictors of burnout and emotional exhaustion across a spectrum of front-line, human service workers as well (e.g., medical doctors, police officers, teachers; Seidler et al. 2014). Teaching seminarians how to deal with these real-world problems is difficult, and preparing them is a more challenging task.

## 2. Seminary Training

Indeed, those training to be clergy (i.e., seminary students) have plenty of responsibilities to attend to, and they often find themselves in a unique subculture that can be difficult to describe to those not familiar with theological and spiritual fields of study. For example, many clergy and seminarians report receiving a "calling" regarding their chosen field of work or study. While people in other professions, especially helping professions, may report a similar magnetism towards a particular field, what sets this experience apart is that the origin is divine (Thomas 1962). Moreover, experiencing a call to vocational ministry is often assumed or presumed. In two separate studies, all participants responded affirmatively to experiencing a call to ministry at some point in their life (Civish 2013; Meek et al. 2003).

Another distinction of seminary training is the aim of developing students in an integrated manner such that they are not only growing in their theological understanding and professional skills, but in their spirituality as well. The terms "spirituality" and "spiritual formation" have an abundance of meanings both within and beyond Christian traditions. A compelling and dynamic conceptualization is that of "relational spirituality" put forth by Shults and Sandage (2006). At its core is a dialectical posture positing movement between a rootedness in a spiritual community and tradition and a healthy curiosity seeking individual expression and intellectual wrestling (Sandage et al. 2011). The impetus of this movement hopefully leads to development along at least three dimensions: spiritual formation (an individual's relationship with God and/or the sacred), characterological formation (internalization of virtue), and moral formation (outward expression of virtue; Porter et al. 2019).

The aspiration of promoting the spiritual growth of seminarians is easier said than done. While interviewing seminarians, Greenman and Siew (2001) reported nearly all subjects in the study alluded to the potential dampening or disruption to their personal spiritual lives because of the rigor of their academic demands. Furthermore, the interviewers solicited comments regarding public engagement as an expression of spirituality which adds yet another dimension to which seminarians may be attempting to embody their vocational calling amidst their academic responsibilities. When helping seminarians thrive while juggling the various demands, it is important for training institutions to be attentive to the holistic health of their students.

## 3. The Mental Health of Seminary Students

While much of the attention has understandably gone towards investigating the spiritual health and development of seminary students, the mental health of seminary students is interesting for a variety of reasons. Perhaps most obvious is mental health as an indicator of their quality of life and well-being which hopefully impacts their ability to thrive during their seminary journey. It is important, first, to pause and acknowledge the nuance between "mental health" and "mental illness." "Mental health" denotes strengths-based terms such as emotional well-being, effective behaviors, meaningful relationships, and the ability to cope with stressors (American Psychological Association n.d.). On the other hand, "mental illness" denotes deficits-based terms such as emotional and cognitive disturbances and corresponding abnormal manifestations and/or functional impairment caused by these symptoms (American Psychological Association n.d.). While the two concepts have been frequently conflated, recent studies confirm that they are indeed distinct yet correlated axes rather than poles of a single continuum (Keyes 2005; Payton 2009). Although the variables in the present study focus primarily on mental illness, the results of this study will be considered from a larger perspective that includes ideas pertaining to mental health as well.

Unfortunately, there is a dearth of research focusing on the mental health of seminarians. Moreover, within this small body of literature, disparate results illuminate the breadth of trajectories future investigation may take. For example, one study found that seminarians reported higher quality of life than their secular peers in psychophysical, psychosocial,

personal, and metaphysical spheres (Prusak et al. 2021), while another study found that the-ology students and pastors scored lower on measures of mental health compared to norm samples (Voltmer et al. 2011). The former study attributed their results to the numerous studies validating the positive impact of religiosity and spirituality on well-being in the way of social support, purpose in life, and positive orientation. For example, one longitudinal study reported a longitudinal relationship between religious/spiritual commitment and well-being that was mediated by virtue (Jankowski et al. 2022a). On the other hand, the second study stressed the psychosocial strain incurred in helping professions such as voca-tional ministry. Similar to the distinction made between mental health and mental illness (Keyes 2005; Payton 2009), the former study employed measures that generally map onto mental health (e.g., gratitude, meaning in life, psychological quality in life), while the latter used instruments that map onto mental illness (e.g., mental health symptoms, physical health impairments). Thus, while the results appear to be contradictory at the surface level, these divergent results are not necessarily orthogonal in nature. The pain/strain incurred in vocational ministry may be heavy, but it may not have to reduce the felt meaning, purpose, and quality of life of seminary students.

Despite being several decades old, an article by the Rev. Dr. Owen Thomas (1962) provides insight into the psychological strain that may still resonate with present-day seminarians. First, he points out that the content of seminary study is both "deeply personal and existential," covering topics such as ultimate destiny, human nature, and redemption. Second, he identifies the "irrational anomaly" of the calling experience (see Seminary Training section). Finally, because of this calling, doubts about their vocation can become overbearing, and the pressure to finish seminary is intensified by the shame at the thought of "failing" their call. These among others are unique psychological stressors that seminarians may face.

It is also important to recognize contextual differences and individual needs. For example, Catholic seminarians in Poland report minimal anxiety regarding their career trajectory since this decision would be made for them (Prusak et al. 2021), whereas Catholic seminarians in Nigeria expressed a greater desire for more career counseling given the lack of guidance in their context (Adubale and Aluede 2017). Another specific expressed need surfacing in the Nigeria study was for financial counseling. Financial burden is a reality for many seminarians given the rising costs of higher education and the relatively low earning potential of their chosen field; however, this goes beyond material pragmatics. One study found that the negative relationship between financial well-being and mental illness, in the form of depression and anxiety, was mediated by shame (Blea et al. 2021). While the precise reason for this relationship requires further investigation, it is clear that the stresses that seminarians face can have a deeply personal impact and contribute to mental illness.

While further research is needed to better understand mental health and illness among seminarians, it is undoubtedly important for seminarians to develop coping skills and healthy practices given the prevalence of mental illness among clergy. Multiple studies have found increased rates of depression and anxiety among clergy members compared to the general population (Proeschold-Bell et al. 2013; Shaw et al. 2021), and higher levels of mental illness have been associated with social isolation, loneliness, and critical and demanding congregations (Proeschold-Bell et al. 2015; Terry and Cunningham 2020). A longitudinal study by Milstein et al. (2020) helped to clarify the directionality of these interactions in which greater spiritual well-being predicts lower depressive symptoms, which in turn predicts less occupational distress. They note that this direction suggests that while occupational stressors may not necessarily cause depression, depressive symptoms may increase susceptibility to occupational distress.

The need to help seminarians promote healthy practices and reduce mental illness risk factors has only become more pronounced due to the increased strain from the COVID-19 pandemic. Research presented by Scott McConnell investigating pastor attrition in the Fall of 2021 (personal communication, 11 February 2022) found that 50% of pastors feel that the demands of ministry are greater than they can handle. The proportion of pastors who feel

"frequently overwhelmed" was recorded at 63%, up from 54% in 2015. Corresponding with other findings regarding pastoral social support, 38% of pastors reported feeling isolated in their role. These numbers corroborate findings from Barna who found that as of March 2022, 42% of pastors considered quitting full-time ministry within the past year (Barna Group 2022). While some metrics improved, such as pastors' sense of financial security and their feeling of being "on call", continued efforts are needed to increase clergy resiliency.

## 4. Prayer Practices and Types

Many occupations place professionals in the unenviable limelight to "practice what they preach," lest they be labeled a hypocrite; however, few may feel this pressure more than clergy for whom this maxim has a quite literal meaning. Prayer, defined as a personal or corporate act of communication between humans and the divine (e.g., God, the gods, supernatural entities; Illueca and Doolittle 2020), is a ubiquitous spiritual practice across many religions and spiritual traditions, and for individuals who seek to lead their communities in prayer practices, it is reasonably expected that clergy develop their own practice of this discipline.

Although studies on prayer have begun to pervade the scientific literature, it remains a difficult phenomenon to study. Methodologically, many previous studies have suffered from a lack of rigor including design flaws, weak measurement tools, and internal validity problems (McCullough 1995). From a theological perspective, the primary goal of prayer is relating to God, which is often not at the center of the research question. While this need not necessarily exclude the study of prayer, taking into consideration cultural diversity and inclusion, it behooves researchers to acknowledge the religious and spiritual values at play. Thus, studies examining prayer practices may focus on secondary goals and tangential effects. A review of empirical research on the relationship between prayer and health tends to divide into the following domains: prayer and subjective well-being, prayer as coping, the effects of intercessory prayer, content of prayer, and prayer and psychiatric symptoms/disorders (Masters and Spielmans 2007; McCullough 1995). Our study will focus on prayer and psychiatric symptoms/disorders.

When examining the relationship between prayer and psychiatric symptoms, we will consider prayer from a multidimensional perspective. The effects of prayer on psychiatric symptoms/disorders appear to be dependent on one's image about God (e.g., benevolent versus vindictive God; Bradshaw et al. 2008), attachment style to God (e.g., avoidance versus secure attachment; Ellison et al. 2014), type of prayer (e.g., meditative versus petitionary prayers; Poloma and Pendleton 1989), increased control through God (Jeppsen et al. 2015), and closeness to God (Jeppsen et al. 2015). With regards to private prayer, Laird et al. (2004) divided prayer into four types: adoration (i.e., focus on worship of God), confession (i.e., acknowledging own shortcomings), thanksgiving (i.e., expressions of gratitude), and supplication (i.e., requests for God to intervene). Covering a broader spectrum of prayers (beyond private prayers), Poloma and Pendleton (1989) identified four types of prayers: petitionary, colloquial, ritualistic, and meditative prayer types based on principal components analysis and demonstrated theorized relationships with external measures of quality of life. Petitionary prayer refers to requests for specific material needs of self and others. Colloquial prayer includes talking to God in one's own words, generally being less concrete and specific compared to petitionary prayers. Ritualistic prayer refers to prayers that are read, memorized, and recited. Meditative prayer includes components of intimacy and personal relationship to God such as adoring, reflecting, and communicating with the divine.

Poloma and Pendleton (1989, 1991) reported that the content and intent of the prayer type being practiced makes a difference. In their study on the effects of prayer types on general well-being, they found that frequency of prayer was a weak predictor of well-being and that experiences during prayer and types of prayer practices engaged in were stronger predictors of well-being (1991). Furthermore, they found that petitionary prayer was the only type not related to any measure of quality of life; meditative, colloquial, and ritual

prayer were related to at least one specific subcategory of quality of life (1989). In fact, petitionary prayer has been found to be consistently associated with negative mental health outcomes (Black et al. 2015; Jeppsen et al. 2015; Zarzycka and Krok 2021). Upon closer investigation, several studies have shown that prayer practices predict enhanced mental health and well-being through various mechanisms including closeness to God and feeling like God is in control (Jeppsen et al. 2015), disclosure as a means of communicating with God (Black et al. 2015; Zarzycka and Krok 2021); trust-based beliefs (Pössel et al. 2014), and cognitive reappraisal (i.e., reframing the source of pain; Dezutter et al. 2011). These run parallel to many of the studies reviewed by McCullough (1995) that examined prayer as a coping resource for reducing existing distress, pain, and stress.

## 5. Experiential Avoidance

So how does prayer move from practice to discipline to character growth? Returning to the model of spiritual, characterological, and moral formation set forth by Porter and his colleagues (2019), there is an intermingling of experiencing the Divine, being shaped in a virtuous direction, and expressing virtues in tangible manifestations. They go on to posit that there is value to investigating both the end-result of a formative process as well as the pathways towards those end-results. In this way, the spiritual and character formation of seminarians is an active process requiring intentional, value-driven, and flexible engagement practices that facilitate virtue development within the context of both religious commitment and religious exploration (Bond et al. 2011; Jankowski et al. 2021).

Psychological flexibility in this context enables full contact with the present moment and persisting and changing behavior in service of spiritual character formation (Bond et al. 2011). Conversely, psychological inflexibility occurs when people avoid making contact with private events[1] due to rigid dominance of psychological reactions over chosen values and contingencies. Experiential avoidance is an instance of psychological inflexibility; specifically, it occurs when an individual attempts to alter, escape from, or control difficult, uncomfortable, or undesirable internal states (thoughts, feelings, sensations, etc.), even when this leads to or would lead to actions that are incongruent with one's values and goals (Hayes et al. 2006). Thus, experiential avoidance is diametrically opposed to the notion of psychological flexibility, which is frequently described in the acceptance and commitment therapy model for mental illness (Hayes et al. 1999).

Experiential avoidance has often been shown to only create or exacerbate problems or, ironically, increase interaction with the situation trying to be avoided (Gámez et al. 2011). A review completed by Chawla and Ostafin (2007) of studies investigating the relationship between experiential avoidance and psychopathology found that experiential avoidance was associated with substance use relapse, trauma symptoms, maladaptive coping, poor self-regulation, and psychiatric symptom severity. Conversely, facing the difficulty with acceptance and developing cognitive flexibility have been shown to promote health and work productivity (Bond et al. 2010).

This is congruent with both Christian theology and studies examining faith maturity. Both Jesus and the biblical authors state that life will be filled with difficulty and suffering, especially for those who choose to follow Christ (Matthew 16:24; John 15:18–20; 2 Corinthians 12:10). A cross-sectional study reported by Knabb and Grigorian-Routon (2014) found that those with greater faith maturity tended to utilize positive religious coping strategies such as reaching out to God for support, asking God to help with feelings of anger, and collaborating with God to take action in the face of adversity. They also found that the relationship between negative religious coping strategies (i.e., questioning God's power, viewing suffering as punishment from God) and depression and anxiety was partially mediated by experiential avoidance. Spiritual practices that promote acceptance and psychological flexibility will lead to positive outcomes such as enhanced mental health, reduced mental illness, and higher levels of maturity and well-being.

### 6. The Present Study

Because of the centrality of prayer to Christian spirituality, it is not difficult to see how prayer sits squarely in the spiritual formation domain as a primary means to relate to God and submit to God's will. Thus, this study will examine the downstream effects of specific types of prayers within Porter et al.'s (2019) spiritual and characterological development framework. Specifically, this study will shed light on the longitudinal direct and indirect pathways in which prayer may protect against or predispose toward experiencing mental illness. To this end, experiential avoidance will be examined longitudinally as a means to track the development of personal virtues resulting from prayer activities. Moreover, because previous studies investigating prayer operationalize mental health as a general, broad construct referred to as overall mental health (Black et al. 2015; Jeppsen et al. 2015), we have chosen to examine the particular relationships between prayer types and depression, anxiety, and burnout symptoms to provide greater specificity in the literature.

To these ends, we submit the following hypotheses. First, following the example of Poloma and Pendleton (1989, 1991), we hypothesize that the total frequency of prayer will be a weak predictor of mental illness, and specificity of prayer types will be a greater indicator of mental health. Second, we hypothesize that colloquial, liturgical (ritual), and meditative prayer will be negatively related with depression, anxiety, and burnout symptoms. Given the frequent association of petitionary prayer with negative mental health outcomes, we hypothesize that petitionary prayer will positively predict depression, anxiety, and burnout symptoms. Third, building off Hayes and his colleague's (1999) conceptualization of experiential avoidance, we hypothesize that experiential avoidance will positively predict depression, anxiety, and burnout symptoms. Fourth, we hypothesize that colloquial, liturgical, and meditative prayer will negatively predict experiential avoidance, and petitionary prayer will positively predict experiential avoidance. Fifth, for our primary model of interest, we hypothesize that experiential avoidance will mediate the relationship between each of the four prayer types and each of the negative mental health outcomes. Although measuring the impact of the COVID-19 pandemic was not the primary aim of this study when data was initially collected in the fall of 2019, the longitudinal nature of this project provides additional context for the variables of interest as they apply to this contemporary reality.

### 7. Method

#### 7.1. Participants

Data were collected from students at 17 seminaries across North America (N = 564; Mage = 31.49; SD = 11.12; range = 19–71). Participants included non-degree, graduate degree (e.g., M.A., M.Div.), or certificate-earning students. Among those providing demographic data, 52.0% identified as male, 47.3% identified as female, and 0.7% identified as other or chose not to disclose. The majority of the sample identified as White (62.9%), followed by Asian (14.3%), Black (8.7%), Hispanic (6.0%), multiracial (6.1%), and other (2.0%).

#### 7.2. Procedure

Upon receiving Institutional Review Board approval, first-term students were recruited during the fall of 2019 via digital and in-person announcements. Participants provided voluntary consent and completed an online survey containing a variety of measures discussed below. Students were compensated with a USD 25 gift card for their participation and were invited in subsequent semesters approximately six months apart to complete the survey again. Accordingly, participants completed time point 1 measures in the fall of 2019, time point 2 measures in the spring of 2020, time point 3 measures in the fall of 2020, and time point 4 measures in the spring of 2021.

## 8. Measures

### 8.1. Prayer Types

Frequency engaging in various prayer practices was measured using a scale consisting of eight items with two items for each prayer type, resulting in four subscales (colloquial, liturgical, meditative, and petitionary). This scale was based on the factor analysis of 15 prayer activity items formulated by Poloma and Pendleton (1989) in which they identified four factors ("types of prayer"). "Ritual" prayer was reconceptualized as "liturgical" prayer. Items are rated on a 5-point Likert-type scale ranging from 1 (*never*) to 5 (*very often*), with higher scores indicating more frequent practice of the prayer activity associated with the prayer type and lower scores indicating less frequency. Scores of the two items per subscale were aggregated to produce a single score for each prayer type. Examples of statements that participants were prompted to respond to included: "Talk with God as you talk with a friend" (colloquial), "Recite prayers that you have memorized" (liturgical), "Spend time being in the presence of God" (meditative), and "Ask God for material things you may need" (petitionary). For total frequency of prayer, we did not have a separate item as Poloma and Pendleton (1989, 1991) did and instead used an aggregate variable of all eight items.

### 8.2. Experiential Avoidance

Experiential avoidance was measured using items from the Acceptance and Action Questionnaire, version 2 (AAQ-II), a measure designed to measure experiential avoidance and psychological inflexibility (Bond et al. 2011). Although their factor analysis resulted in a unidimensional measure centered around the construct of psychological inflexibility, experiential avoidance is a foundational concept within this construct and reflected in several of the items. For the purposes of the present study, we used three items that most closely reflect the concept of experiential avoidance and had high factor loadings: "I'm afraid of my feelings" (0.70), "I worry about not being able to control my worries and feelings" (0.71), and "Emotions cause problems in my life" (0.79). Items were rated on a 7-point Likert-type scale ranging from 1 (*never true*) to 7 (*always true*), and responses were aggregated to form a single latent concept.

### 8.3. Depression

Depression symptoms were measured using the Patient Health Questionnaire-9 (PHQ-9), a diagnostically validated, self-report screening measure frequently used in clinical settings (Kroenke et al. 2001; Spitzer et al. 1999). It consists of nine items which ask about the frequency of depressive symptoms within the past two weeks. Responses for each item are rated on a 4-point Likert-type scale ranging from 0 to 3, with a possible overall scale range of 0 to 27. Higher scores indicate higher levels of impairment with cutoffs for clinical use being: 1–4, minimal depression; 5–9, mild depression; 10–14, moderate depression; 15–19, moderately severe depression; and 20–27, severe depression.

### 8.4. Anxiety

Anxiety symptoms were measured using the Generalized Anxiety Disorder Scale-7 (GAD-7), a diagnostically validated, self-report screening measure frequently used in clinical settings (Spitzer et al. 2006). It consists of seven items which ask about the frequency of core symptoms of generalized anxiety disorder within the past two weeks. Responses for each item are rated on a 4-point Likert-type scale ranging from 0 to 3, with a possible overall scale range of 0 to 21. Higher scores indicate higher levels of impairment with cutoffs for clinical use being: 1–5, mild anxiety; 6–10, moderate anxiety; and 15–21, severe anxiety.

### 8.5. Work-Related Burnout

Work-related burnout was measured using the corresponding scale of the Copenhagen Burnout Inventory (CBI; Kristensen et al. 2005). The CBI was developed to have three sub-dimensions (personal burnout, work-related burnout, and client related burnout) that

could be used independently or integratively. The developers of this scale cite theoretical and methodological reasons for choosing not to validate the three CBI scales through factor analysis, one such reason being that the scales may be used to fit with the populations being studied. They define work-related burnout as "the degree of physical and psychological fatigue and exhaustion that is perceived by the person as related to his/her work" (Kristensen et al. 2005). The work-related burnout subscale consists of seven items rated on a 5-point Likert-type scale ranging from 1 (*never/almost never*) to 5 (*always/almost always*). In the validation study, scoring ranged from 0 (*never/almost* never) to 100 (*always/almost* always). Example items include "Is your work emotionally exhausting?" and "Do you feel worn out at the end of the working day?"

## 9. Statistical Analyses

Data analysis utilized the Statistical Package for the Social Sciences (SPSS) version 28. We tested our hypotheses using bivariate and multiple regression models and utilized Hayes's (2018) PROCESS macro for SPSS Model 4 to test a series of simple mediation models with covariates. Data from four time points were analyzed to investigate the longitudinal relationship between variables. In addition to the proposed model of experiential avoidance mediating the relationship between each prayer type and each negative mental health outcome, we tested three other models to confirm the directionality of the relationships. In the proposed model, the independent variables were each of the four prayer types at Time 1, each tested separately. The dependent variables were depression and anxiety at Time 3, each tested separately. Burnout was also tested as a dependent variable; however, data for burnout was only available at Time 4. Experiential avoidance was measured at Time 2 as the mediator variable.

The following are the other models tested to confirm the directionality of the relationships and to test for better model fits. Model 2 used experiential avoidance at Time 1 as the dependent variable, each prayer type at Time 1 used separately as the mediator variable, and depression and anxiety at Time 3 and burnout at Time 4 used separately as the dependent variable. Model 3 used experiential avoidance at Time 1 as the dependent variable, each prayer type at Time 2 used separately as the mediator variable, and depression and anxiety at Time 3 and burnout at Time 4 used separately as the dependent variable. Model 4 used experiential avoidance at Time 1 as the dependent variable, all prayer type at Time 2 used as parallel mediators, and depression and anxiety at Time 3 and burnout at Time 4 used separately as the dependent variable.

## 10. Results

Means and standard deviations for the variables studied are displayed in Table 1. The mean score for the PHQ-9 in the present study (M = 5.87, SD = 5.15) was comparable to some normative data. A study in Germany found that in non-clinical settings, women had mean scores of 3.1 (*SD* = 3.5) and men had mean scores of 2.7 (*SD* = 3.5; Kocalevent et al. 2013); however, a study in Japan found mean scores of 6.96 (*SD* = 6.46) among a non-clinical group and mean scores of 12.42 (*SD* = 7.57) among those diagnosed with major depressive disorder (Doi et al. 2018). Some 19 percent of participants scored 10 or higher, which is regarded as the cutoff score with the highest combined sensitivity and specificity for major depressive disorder (Levis et al. 2019). The mean score for the GAD-7 in the present study (*M* = 4.93, *SD* = 4.71) was likewise very similar to previous findings of mean scores among a general population (*M* = 4.9, *SD* = 4.8; Spitzer et al. 2006). Some 16 percent of participants scored 10 or higher which is the cutoff for a potential diagnosis of generalized anxiety disorder.

**Table 1.** Means and standard deviations of variables tested in regression models.

|  | *M* | *SD* |
|---|---|---|
| Petitionary Prayer | 3.06 | 0.92 |
| Liturgical Prayer | 2.75 | 1.21 |
| Meditative Prayer | 3.83 | 0.90 |
| Colloquial Prayer | 3.87 | 0.95 |
| Experiential Avoidance | 2.97 | 1.38 |
| Anxiety Symptoms | 4.93 | 4.71 |
| Depression Symptoms | 5.87 | 5.15 |
| Work Burnout | 18.37 | 5.15 |

Table 2 displays the correlation matrix as a preliminary overview for the variables studied in the proposed model. As expected, the majority of the prayer types correlated with each other as did all three of the negative mental health outcomes. Petitionary prayer and liturgical prayer did not correlate with each other. Interestingly, depression symptoms had a negative relationship with both liturgical prayer ($r = -0.11$; small effect, $p < 0.05$) and meditative prayer ($r = -0.14$; small effect, $p < 0.01$).

**Table 2.** Correlations among prayer types, experiential avoidance, and negative mental health.

|  | 1 | 2 | 3 | 4 | 5 | 6 | 7 | 8 |
|---|---|---|---|---|---|---|---|---|
| 1. Petitionary Prayer | - |  |  |  |  |  |  |  |
| 2. Liturgical Prayer | 0.058 | - |  |  |  |  |  |  |
| 3. Meditative Prayer | 0.118 ** | 0.293 ** | - |  |  |  |  |  |
| 4. Colloquial Prayer | 0.276 ** | 0.109 ** | 0.517 ** | - |  |  |  |  |
| 5. Experiential Avoidance | −0.024 | −0.138 ** | −0.146 ** | −0.165 ** | - |  |  |  |
| 6. Anxiety Symptoms | 0.029 | −0.079 | −0.095 | −0.081 | 0.345 ** | - |  |  |
| 7. Depression Symptoms | −0.078 | −0.113 * | −0.144 ** | −0.093 | 0.401 ** | 0.709 ** | - |  |
| 8. Work Burnout | 0.079 | −0.051 | −0.074 | −0.086 | 0.275 ** | 0.405 ** | 0.454 ** | - |

* $p < 0.05$; ** $p < 0.01$.

Our first hypothesis predicted that the total prayer score would demonstrate a weak relationship with anxiety, depression, and burnout. The standardized direct effects on anxiety ($\beta = 0.01$, $p = 0.93$), depression ($\beta = -0.06$, $p = 0.25$), and work burnout ($\beta = 0.01$, $p = 0.77$) were all insignificant. Interestingly, when we added the mediator variable of experiential avoidance as in the primary model of interest into the regression model, the results were significant: anxiety (Table 3, standardized indirect effect = −0.08, 95% CI [−0.13, −0.03]); depression (Table 3, standardized indirect effect = −0.72, 95% CI [−1.19, −0.35]); and work burnout (Table 3, standardized indirect effect = −0.55, 95% CI [−0.94, −0.24]). These results partially support our hypothesis.

Our second hypothesis predicted that colloquial, liturgical, and meditative prayer would negatively predict anxiety, depression, and burnout and that petitionary prayer would positively predict depression, anxiety, and burnout. For colloquial prayer, the standardized direct effects on anxiety ($\beta = 0.01$, $p = 0.80$), depression ($\beta = -0.01$, $p = 0.83$), and work burnout ($\beta = -0.01$, $p = 0.82$) were all insignificant. The standardized direct effects of liturgical prayer on anxiety ($\beta = -0.05$, $p = 0.37$), depression ($\beta = -0.04$, $p = 0.50$), and work burnout ($\beta = -0.01$, $p = 0.89$) were all insignificant. For meditative prayer, the standardized direct effects on anxiety ($\beta = -0.004$, $p = 0.94$), depression ($\beta = -0.06$, $p = 0.25$), and work burnout ($\beta = 0.01$, $p = 0.93$) were all insignificant. The direct effects between petitionary prayer and anxiety ($\beta = 0.06$, $p = 0.20$), depression ($\beta = -0.08$, $p = 0.16$), and work burnout ($\beta = 0.10$, $p = 0.07$) were all insignificant. These results did not support our hypothesis.

**Table 3.** Regression models involving total prayer, experiential avoidance and mental health symptoms.

| | | Consequent Variable | | | | | | |
|---|---|---|---|---|---|---|---|---|
| | | (M) Experiential Avoidance | | | | (Y) Anxiety | | |
| Antecedent variable | | Coeff | *SE* | *p* | | Coeff | *SE* | *p* |
| (X) Total Prayer | $a_1$ | −0.49 | 0.12 | 0.000 | $c'_1$ | 0.01 | 0.06 | 0.929 |
| (M) Experiential Avoidance | - | - | - | - | $b_1$ | 0.16 | 0.02 | 0.001 |
| Constant | $I_{M1}$ | 4.82 | 0.43 | 0.000 | $I_{Y1}$ | 0.44 | 0.24 | 0.071 |
| | | $R^2 = 0.050$ | | | | $R^2 = 0.13$ | | |
| | | $F_{(2, 333)} = 8.68, p < 0.01$ | | | | $F_{(3, 332)} = 17.07, p < 0.01$ | | |
| | | Consequent Variable | | | | | | |
| | | (M) Experiential Avoidance | | | | (Y) Depression | | |
| Antecedent variable | | Coeff | *SE* | *p* | | Coeff | *SE* | *p* |
| (X) Total Prayer | $a_2$ | −0.49 | 0.12 | 0.000 | $c'_2$ | −0.52 | 0.45 | 0.245 |
| (M) Experiential Avoidance | - | - | - | - | $b_2$ | 1.47 | 0.20 | 0.000 |
| Constant | $i_{M2}$ | 4.82 | 0.43 | 0.000 | $i_{Y2}$ | 203.31 | 1.86 | 0.000 |
| | | $R^2 = 0.050$ | | | | $R^2 = 0.18$ | | |
| | | $F_{(2, 333)} = 8.68, p < 0.01$ | | | | $F_{(3, 332)} = 24.94, p < 0.01$ | | |
| | | Consequent Variable | | | | | | |
| | | (M) Experiential Avoidance | | | | (Y) Work Burnout | | |
| Antecedent variable | | Coeff | *SE* | *p* | | Coeff | *SE* | *p* |
| (X) Total Prayer | $a_3$ | −0.54 | 0.13 | 0.000 | $c'_3$ | 0.13 | 0.47 | 0.770 |
| (M) Experiential Avoidance | - | - | - | - | $b_3$ | 1.02 | 0.21 | 0.000 |
| Constant | $i_{M3}$ | 5.02 | 0.48 | 0.000 | $i_{Y3}$ | 17.01 | 1.94 | 0.000 |
| | | $R^2 = 0.058$ | | | | $R^2 = 0.12$ | | |
| | | $F_{(2, 272)} = 8.41, p < 0.01$ | | | | $F_{(3, 271)} = 11.99, p < 0.01$ | | |

*Note.* Estimates represented in the table are unstandardized parameter estimates. The corresponding standardized indirect effects predicting dependent outcomes as follows: anxiety = −0.08, 95% CI [−0.13, −0.03]; depression = −0.72, 95% CI [−1.19, −0.35]; work burnout = −0.55, 95% CI [−0.94, −0.24].

Our third hypothesis predicted that experiential avoidance will positively predict depression, anxiety, and burnout symptoms. In each of the models tested, experiential avoidance positively predicted anxiety: colloquial (Table 4; $b_1 = 0.16$, $p < 0.01$), liturgical (Table 5; $b_1 = 0.16$, $p < 0.01$), meditative (Table 6; $b_1 = 0.16$, $p < 0.01$), and petitionary (Table 7; $b_1 = 0.16$, $p < 0.01$). Similarly, in each of the models tested, experiential avoidance positively predicted depression: colloquial (Table 4; $b_2 = 1.64$, $p < 0.01$), liturgical (Table 5; $b_2 = 1.62$, $p < 0.01$), meditative (Table 6; $b_2 = 1.60$, $p = 0.01$), and petitionary (Table 7; $b_2 = 1.63$, $p < 0.01$). Finally, in each of the models, experiential avoidance positively predicted work burnout: colloquial (Table 4; $b_3 = 1.04$, $p < 0.01$), liturgical (Table 5; $b_3 = 1.04$; $p < 0.01$), meditative (Table 6; $b_3 = 1.04$; $p < 0.01$), and petitionary (Table 7; $b_3 = 1.06$, $p < 0.01$). Our third hypothesis was fully supported in all models.

**Table 4.** Regression models involving colloquial prayer, experiential avoidance, and mental health symptoms.

| | | Consequent Variable | | | | | | |
| --- | --- | --- | --- | --- | --- | --- | --- | --- |
| | | (M) Experiential Avoidance | | | | (Y) Anxiety | | |
| Antecedent variable | | Coeff | SE | p | | Coeff | SE | p |
| (X) Colloquial Prayer | $a_1$ | −0.24 | 0.08 | 0.002 | $c'_1$ | 0.01 | 0.04 | 0.801 |
| (M) Experiential Avoidance | - | - | - | - | $b_1$ | 0.16 | 0.03 | 0.000 |
| Constant | $I_{M1}$ | 4.06 | 0.33 | 0.000 | $I_{Y1}$ | 0.42 | 0.19 | 0.027 |
| | | $R^2 = 0.03$ $F(2, 333) = 5.00, p < 0.01$ | | | | $R^2 = 0.13$ $F(3, 332) = 17.09, p < 0.01$ | | |
| | | Consequent Variable | | | | | | |
| | | (M) Experiential Avoidance | | | | (Y) Depression | | |
| Antecedent variable | | Coeff | SE | p | | Coeff | SE | p |
| (X) Colloquial Prayer | $a_2$ | −0.30 | 0.09 | 0.001 | $c'_2$ | −0.07 | 0.32 | 0.829 |
| (M) Experiential Avoidance | - | - | - | - | $b_2$ | 1.64 | 0.22 | 0.000 |
| Constant | $i_{M2}$ | 4.07 | 0.39 | 0.000 | $i_{Y2}$ | 200.91 | 1.64 | 0.000 |
| | | $R^2 = 0.04$ $F(2, 272) = 5.73, p < 0.01$ | | | | $R^2 = 0.19$ $F(3, 271) = 21.31, p < 0.01$ | | |
| | | Consequent Variable | | | | | | |
| | | (M) Experiential Avoidance | | | | (Y) Work Burnout | | |
| Antecedent variable | | Coeff | SE | p | | Coeff | SE | p |
| (X) Colloquial Prayer | $a_3$ | −0.32 | 0.09 | 0.000 | $c'_3$ | −0.07 | 0.30 | 0.818 |
| (M) Experiential Avoidance | - | - | - | - | $b_3$ | 1.04 | 0.20 | 0.000 |
| Constant | $i_{M3}$ | 4.19 | 0.37 | 0.000 | $i_{Y3}$ | 18.26 | 1.49 | 0.000 |
| | | $R^2 = 0.01$ $F(2, 293) = 6.77, p < 0.01$ | | | | $R^2 = 0.14$ $F(3, 292) = 15.31, p < 0.01$ | | |

*Note.* Estimates represented in the table are unstandardized parameter estimates. The corresponding standardized indirect effects predicting dependent outcomes as follows: anxiety = −0.04, 95% CI [−0.07, −0.01] depression = −0.49, 95% CI [−0.86, −0.17]; work burnout = −0.33, 95% CI [−0.60, −0.12].

**Table 5.** Regression models involving liturgical prayer, experiential avoidance, and mental health symptoms.

| | | Consequent Variable | | | | | | |
| --- | --- | --- | --- | --- | --- | --- | --- | --- |
| | | (M) Experiential Avoidance | | | | (Y) Anxiety | | |
| Antecedent variable | | Coeff | SE | p | | Coeff | SE | p |
| (X) Liturgical Prayer | $a_1$ | −0.18 | 0.06 | 0.004 | $c'_1$ | −0.03 | 0.03 | 0.365 |
| (M) Experiential Avoidance | - | - | - | - | $b_1$ | 0.16 | 0.03 | 0.000 |
| Constant | $I_{M1}$ | 3.72 | 0.25 | 0.000 | $I_{Y1}$ | 0.55 | 0.15 | 0.000 |
| | | $R^2 = 0.03$ $F(2, 333) = 4.57, p = 0.011$ | | | | $R^2 = 0.14$ $F(3, 332) = 17.38, p < 0.01$ | | |
| | | Consequent Variable | | | | | | |
| | | (M) Experiential Avoidance | | | | (Y) Depression | | |
| Antecedent variable | | Coeff | SE | p | | Coeff | SE | p |
| (X) Liturgical Prayer | $a_2$ | −0.19 | 0.07 | 0.007 | $c'_2$ | −0.17 | 0.25 | 0.499 |
| (M) Experiential Avoidance | - | - | - | - | $b_2$ | 1.62 | 0.21 | 0.000 |
| Constant | $i_{M2}$ | 3.53 | 0.29 | 0.000 | $i_{Y2}$ | 201.19 | 1.28 | 0.000 |
| | | $R^2 = 0.03$ $F(2, 272) = 3.81, p = 0.023$ | | | | $R^2 = 0.19$ $F(3, 271) = 21.48, p < 0.01$ | | |

**Table 5.** *Cont.*

| | | | Consequent Variable | | | | | |
|---|---|---|---|---|---|---|---|---|
| | | (M) Experiential Avoidance | | | | (Y) Work Burnout | | |
| Antecedent variable | | Coeff | SE | p | | Coeff | SE | p |
| (X) Liturgical Prayer | $a_3$ | −0.21 | 0.06 | 0.001 | $c'_3$ | −0.03 | 0.22 | 0.893 |
| (M) Experiential Avoidance | - | - | - | - | $b_3$ | 1.04 | 0.19 | 0.000 |
| Constant | $i_{M3}$ | 3.63 | 0.27 | 0.000 | $i_{Y3}$ | 18.08 | 1.15 | 0.000 |
| | | $R^2 = 0.03$ | | | | $R^2 = 0.14$ | | |
| | | $F(2, 293) = 5.24, p < 0.01$ | | | | $F(3, 292) = 15.29, p < 0.01$ | | |

*Note.* Estimates represented in the table are unstandardized parameter estimates. The corresponding standardized indirect effects predicting dependent outcomes as follows: anxiety = −0.03, 95% CI [−0.05, −0.01]; depression = −0.30, 95% CI [−0.55, −0.08]; work burnout = −0.22, 95% CI [−0.37, −0.09].

**Table 6.** Regression models involving meditative prayer, experiential avoidance, and mental health symptoms.

| | | | Consequent Variable | | | | | |
|---|---|---|---|---|---|---|---|---|
| | | (M) Experiential Avoidance | | | | (Y) Anxiety | | |
| Antecedent variable | | Coeff | SE | p | | Coeff | SE | p |
| (X) Meditative Prayer | $a_1$ | −0.25 | 0.08 | 0.003 | $c'_1$ | −0.003 | 0.04 | 0.937 |
| (M) Experiential Avoidance | - | - | - | - | $b_1$ | 0.16 | 0.03 | 0.000 |
| Constant | $I_{M1}$ | 4.10 | 0.35 | 0.000 | $I_{Y1}$ | 0.47 | 0.19 | 0.017 |
| | | $R^2 = 0.03$ | | | | $R^2 = 0.13$ | | |
| | | $F(2, 333) = 4.85, p < 0.01$ | | | | $F(3, 332) = 17.07, p < 0.01$ | | |

| | | | Consequent Variable | | | | | |
|---|---|---|---|---|---|---|---|---|
| | | (M) Experiential Avoidance | | | | (Y) Depression | | |
| Antecedent variable | | Coeff | SE | p | | Coeff | SE | p |
| (X) Meditative Prayer | $a_2$ | −0.29 | 0.09 | 0.002 | $c'_2$ | −0.38 | 0.33 | 0.246 |
| (M) Experiential Avoidance | - | - | - | - | $b_2$ | 1.60 | 0.21 | 0.000 |
| Constant | $i_{M2}$ | 4.09 | 0.41 | 0.000 | $i_{Y2}$ | 202.22 | 1.68 | 0.000 |
| | | $R^2 = 0.04$ | | | | $R^2 = 0.19$ | | |
| | | $F(2, 272) = 5.28, p < 0.01$ | | | | $F(3, 271) = 21.85, p < 0.01$ | | |

| | | | Consequent Variable | | | | | |
|---|---|---|---|---|---|---|---|---|
| | | (M) Experiential Avoidance | | | | (Y) Work Burnout | | |
| Antecedent variable | | Coeff | SE | p | | Coeff | SE | p |
| (X) Meditative Prayer | $a_3$ | −0.31 | 0.09 | 0.001 | $c'_3$ | −0.03 | 0.30 | 0.929 |
| (M) Experiential Avoidance | - | - | - | - | $b_3$ | 1.04 | 0.20 | 0.000 |
| Constant | $i_{M3}$ | 4.20 | 0.39 | 0.000 | $i_{Y3}$ | 18.09 | 1.54 | 0.000 |
| | | $R^2 = 0.04$ | | | | $R^2 = 0.14$ | | |
| | | $F(2, 293) = 6.10, p < 0.01$ | | | | $F(3, 292) = 15.29, p < 0.01$ | | |

*Note.* Estimates represented in the table are unstandardized parameter estimates. The corresponding standardized indirect effects predicting dependent outcomes as follows: anxiety = −0.04, 95% CI [−0.07, −0.01]; depression = −0.46, 95% CI [−0.82, −0.17]; work burnout = −0.32, 95% CI [−0.55, −0.13].

**Table 7.** Regression models involving petitionary prayer, experiential avoidance, and mental health symptoms.

| | | Consequent Variable | | | | | | |
|---|---|---|---|---|---|---|---|---|
| | | (M) Experiential Avoidance | | | | (Y) Anxiety | | |
| Antecedent variable | | Coeff | SE | p | | Coeff | SE | p |
| (X) Petitionary Prayer | $a_1$ | −0.07 | 0.08 | 0.395 | $c'_1$ | 0.05 | 0.04 | 0.204 |
| (M) Experiential Avoidance | - | - | - | - | $b_1$ | 0.16 | 0.03 | 0.000 |
| Constant | $I_{M1}$ | 3.42 | 0.31 | 0.000 | $I_{Y1}$ | 0.30 | 0.17 | 0.077 |
| | | $R^2 = 0.004$ | | | | $R^2 = 0.14$ | | |
| | | $F(2, 333) = 0.69, p = 0.50$ | | | | $F(3, 332) = 17.69, p < 0.01$ | | |
| | | Consequent Variable | | | | | | |
| | | (M) Experiential Avoidance | | | | (Y) Depression | | |
| Antecedent variable | | Coeff | SE | p | | Coeff | SE | p |
| (X) Petitionary Prayer | $a_2$ | −0.06 | 0.09 | 0.474 | $c'_2$ | −0.46 | 0.33 | 0.161 |
| (M) Experiential Avoidance | - | - | - | - | $b_2$ | 1.63 | 0.21 | 0.000 |
| Constant | $i_{M2}$ | 3.18 | 0.35 | 0.000 | $i_{Y2}$ | 202.01 | 1.39 | 0.000 |
| | | $R^2 = 0.003$ | | | | $R^2 = 0.20$ | | |
| | | $F(2, 272) = 0.46, p = 0.64$ | | | | $F(3, 271) = 22.10, p < 0.01$ | | |
| | | Consequent Variable | | | | | | |
| | | (M) Experiential Avoidance | | | | (Y) Work Burnout | | |
| Antecedent variable | | Coeff | SE | p | | Coeff | SE | p |
| (X) Petitionary Prayer | $a_3$ | −0.06 | 0.09 | 0.532 | $c'_3$ | 0.55 | 0.30 | 0.071 |
| (M) Experiential Avoidance | - | - | - | - | $b_3$ | 1.06 | 0.19 | 0.000 |
| Constant | $i_{M3}$ | 3.20 | 0.34 | 0.000 | $i_{Y3}$ | 16.37 | 1.26 | 0.000 |
| | | $R^2 = 0.002$ | | | | $R^2 = 0.14$ | | |
| | | $F(2, 293) = 0.22, p = 0.80$ | | | | $F(3, 292) = 16.55, p < 0.01$ | | |

*Note.* Estimates represented in the table are unstandardized parameter estimates. The corresponding standardized indirect effects predicting dependent outcomes as follows: anxiety = −0.01, 95% CI [−0.04, 0.02]; depression = −0.11, 95% CI [−0.46, 0.21]; work burnout = −0.06, 95% CI [−0.29, 0.14].

Our fourth hypothesis predicted that colloquial, liturgical, and meditative prayer would negatively predict experiential avoidance, and petitionary prayer would positively predict experiential avoidance. For all three negative mental health outcomes, colloquial prayer negatively predicted experiential avoidance: anxiety (Table 4; $a_1 = −0.24$, $p < 0.01$), depression (Table 4; $a_2 = −0.30$, $p < 0.01$), and work burnout (Table 4; $a_3 = −0.32$, $p < 0.01$). Liturgical prayer also negatively predicted experiential avoidance for all three negative mental health outcomes: anxiety (Table 5; $a_1 = −0.18$, $p < 0.01$), depression (Table 5; $a_2 = −0.19$, $p < 0.01$), and work burnout (Table 5; $a_3 = −0.21$, $p < 0.01$). Likewise, meditative prayer negatively predicted experiential avoidance for all three negative mental health outcomes: anxiety (Table 6; $a_1 = −0.25$, $p < 0.01$), depression (Table 6; $a_2 = −0.29$, $p < 0.01$), and work burnout (Table 6; $a_3 = −0.31$, $p < 0.01$). In the case of petitionary prayer, its effect on experiential avoidance was insignificant for all three negative mental health outcomes: anxiety (Table 7; $a_1 = −0.07$, $p = 0.40$), depression (Table 7; $a_2 = −0.06$, $p = 0.47$), and work burnout (Table 7; $a_3 = −0.06$, $p = 0.53$). These results mostly supported our hypothesis with only petitionary prayer failing to reach a significant level.

Finally, our fifth hypothesis predicted that experiential avoidance would mediate the relationship between each of the four prayer types and each of the negative mental health outcomes. For the first set of models, experiential avoidance significantly mediated the relationship between colloquial prayer and anxiety (Table 4; standardized indirect effect = −0.04, 95% CI [−0.07, −0.01]), depression (Table 4; standardized indirect effect = −0.49, 95% CI [−0.86, −0.17]), and work burnout (Table 4; standardized indirect effect = −0.33, 95% CI [−0.60, −0.12]). Because the direct effects of colloquial prayer on

all mental health outcomes were not statistically significant (see $c'_1$, $c'_2$, and $c'_3$ in Table 4), experiential avoidance fully mediated the aforementioned relationships, suggesting that anxiety, depression and work burnout symptoms are influenced by engagement with colloquial prayer through a causal sequence in which greater engagement with colloquial prayer leads to less experiential avoidance, which in turn, putatively decreases anxiety, depression, and burnout symptoms.

Similarly, experiential avoidance fully mediated the relationship between liturgical prayer and anxiety (Table 5; standardized indirect effect = −0.03, 95% CI [−0.05, −0.01]), depression (Table 5; standardized indirect effect = −0.30, 95% CI [−0.55, −0.08]), and work burnout (Table 5; standardized indirect effect = −0.22, 95% CI [−0.37, −0.09]). Because the direct effects of liturgical prayer on all three negative mental health outcomes were not statistically significant (see $c'_1$, $c'_2$, and $c'_3$ in Table 5), experiential avoidance fully mediated the aforementioned relationships, suggesting that anxiety, depression, and work burnout symptoms are influenced by engagement with liturgical prayer through a causal sequence in which greater engagement with liturgical prayer leads to less experiential avoidance, which in turn, putatively decreases anxiety, depression, and burnout symptoms.

For meditative prayer, experiential avoidance also fully mediated the relationship with anxiety (Table 6; standardized indirect effect = −0.04, 95% CI [−0.07, −0.01]), depression (Table 6; standardized indirect effect = −0.46, 95% CI [−0.82, −0.17]), and experiential avoidance (Table 6; standardized indirect effect = −0.32, 95% CI [−0.55, −0.13]). Because the direct effects of meditative prayer on all three negative mental health outcomes were not statistically significant (see $c'_1$, $c'_2$, and $c'_3$ in Table 6), experiential avoidance fully mediated the aforementioned relationships, suggesting that anxiety, depression, and work burnout symptoms are influenced by engagement with meditative prayer through a causal sequence in which greater engagement with meditative prayer leads to less experiential avoidance, which in turn, putatively decreases anxiety, depression, and burnout symptoms.

Finally, in the case of petitionary prayer, we did not find a significant mediation effect for anxiety (Table 7 standardized indirect effect = −0.01, 95% CI [−0.04, 0.02]), depression (Table 7; standardized indirect effect = −0.11, 95% CI [−0.46, 0.21]), or work burnout (Table 7; standardized indirect effect = −0.06, 95% CI [−0.29, 0.14]). Because the direct effects between petitionary prayer on all three negative mental health outcomes were not statistically significant as well (see $c'_1$, $c'_2$, and $c'_3$ in Table 7), greater engagement with petitionary prayer appears to have no direct or indirect downstream effects on experiential avoidance of anxiety, depression, or work burnout symptoms.

## 11. Discussion

The aim of the present study was to determine the relationship between different types of prayer, experiential avoidance, and negative mental health outcomes among seminary students. Following the lead of Poloma and Pendleton (1989, 1991), our first test was to examine the overall effect of prayer frequency on mental illness. Similar to their findings, no significant direct effect between total prayer frequency and anxiety, depression, or work burnout was detected. However, when experiential avoidance was added as a mediator, the relationship between total prayer frequency and mental illness became significant. It should be noted that we did not use a separate variable for prayer frequency as did Poloma and Pendleton which may mean we examined slightly different constructs. Nevertheless, the significance of these results bodes well for understanding the longitudinal effects of prayer on mental illness. Our subsequent hypotheses delved into the specificity of individual prayer types.

The second goal was to replicate previous findings linking prayer types with mental health outcomes. While we did find significant indirect effects between certain prayer types and negative mental health, we did not find a significant direct effect as previous studies have found (Black et al. 2015; Jeppsen et al. 2015, 2022). One possibility for why this occurred is that our study used instruments for particular mental health ailments (i.e., GAD-7, PHQ-9, CBI) rather than a general measure for overall mental health. It may

be that identifying such a specific relationship is beyond the scope of our measures, if such a relationship exists at all. Additionally, our study utilized longitudinal data rather than a cross-sectional approach which may weaken the direct effect that prayer practices have on mental states many months into the future. Although Park et al. (2018) did find statistically significant predictive relationships between positive and negative religious coping and well-being in a longitudinal study, religious coping also encompasses a broader range of coping styles beyond prayer, including but not limited to reappraisals of the stressful situation, seeking control through individual initiatives, seeking support from others, and/or religious helping (Pargament et al. 2001).

The third hypothesis that we pursued validated the conceptualization set forth by Hayes and his colleagues (1999) that experiential avoidance leads to negative mental health outcomes. This was fully supported in all the models that were tested. These results confirm the notion that attempting to avoid uncomfortable or difficult situations and feelings serves to exacerbate the challenge of managing mental ailments. One explanation involves conceptualizing emotional avoidance as an emotional discordant stress response (Leonidou and Panayiotou 2021). Specifically, individuals high in dispositional experiential avoidance appear to experience a more intense perception of subjective arousal (controlling for objective physiological activation) during emotional exposures *and* cope by not being fully aware of their responses or the significance of these types of private events. This inaccurate assessment of stress may lead to a mismatch between coping strategies and the type and severity of stressors. Furthermore, when people mitigate or escape difficulty through avoidance, they deny themselves the opportunity to develop the psychological skills and fortitude to grow more capable and resilient. The cost of experiential avoidance is not only that the problem is postponed, but also that we choose incongruent and inauthentic replacements for our true goals and values. These physiological, phenomenological, and behavioral mechanisms represent possible pathways in which experiential avoidance leads to psychopathology (Hayes et al. 1996).

The next hypothesis tested was that colloquial, liturgical, and meditative prayer would negatively predict experiential avoidance and that petitionary prayer would positively predict experiential avoidance. This was mostly supported by our results with petitionary prayer being the only model that was not significant. Our interpretation is that colloquial, liturgical, and meditative prayer help to remain engaged with the present moment in the face of adversity. We posit that petitionary prayer, while it is still a widely encouraged practice in Scripture, can inadvertently lead to experiential avoidance if it is utilized habitually as a wishful attempt to escape. Petitionary prayer has been deemed a less mature form of prayer by some (Black et al. 2015; Poloma and Gallup 1991), and in previous studies, it has been frequently connected with poor mental health outcomes (Black et al. 2015; Jeppsen et al. 2015; Zarzycka and Krok 2021). While our results did not demonstrate a positive relationship between petitionary prayer and experiential avoidance, the lack of significance indicates that more investigation needs to be carried out to illuminate the virtues and pitfalls of this particular practice. Furthermore, future studies should continue to explore directionality to differentiate from individuals whose mental illness may predispose them to utilize petitionary prayer in response to their situation.

Our primary investigatory question was exploring the possibility of experiential avoidance mediating the relationship between prayer types and negative mental health. As with our fourth hypothesis, the models for colloquial, liturgical, and meditative prayer were all significant, and the models for petitionary prayer were insignificant. Thus, colloquial, liturgical, and meditative prayer appears to predict lower experiential avoidance over time, which putatively downregulates anxiety, depression, and burnout symptoms later in life. Additionally, the significance of the mediated model of experiential avoidance on total prayer frequency and mental illness suggests that greater engagement through prayer has a positive effect over time. The fact that the direct effects of all four types of prayer and total prayer on negative mental health were insignificant may help to interpret the longitudinal nature of this relationship. We theorize that the significance of the indirect relationship

over the various time points is indicative of the characterological growth that can occur through these prayer practices which can aid in remaining engaged in the midst of struggle. This perseverance may be due to the intentionality of being disciplined in these practices, the buffering effects of receiving divine or social support, both, or even something more. Previous studies have consistently demonstrated the positive impact of social support for pastors and seminarians, and specifically colloquial and meditative prayer types have been theorized to tap into those relational needs in the human–divine relationship (Black et al. 2015; Jeppsen et al. 2015).

While it is convenient to group the types of prayer together based on the model results or previous findings, it is worth speculating on their individual relationships to experiential avoidance and mental health. Although colloquial prayer may feel uncomfortably informal to some, it is precisely that lack of pretense that allows a person to present their current situation to God in an authentic way. Theologically speaking, there is nothing about us that is hidden from God and trying to hide behind pretense or formality is just another form of experiential avoidance. Moreover, communicating with God in a colloquial manner promotes relational intimacy. Jesus himself encouraged his followers to address his Father as their own, which is an invitation to bring everything before God with a sense of freedom (Matthew 6:9).

Liturgical prayer may be more or less familiar depending on a person's tradition and background, and it can include a variety of practices including repeating a well-known prayer (e.g., the Lord's prayer), reading from a book of prayers, and praying a memorized prayer. In their study of prayer type and mental health outcomes, Black and her colleagues (2015) hypothesized and found that ritual prayer was not significantly related to mental health through self-disclosure as a mediator. Although this makes sense given their theoretical approach, prayer is a multi-layered activity that goes beyond just communicating with God, though that is a primary purpose. Thus, our decision to reframe "ritual" prayer as "liturgical" helps to reflect the nuances and impact of this prayer type, even if those words may be interchangeable in some situations. While there is indeed a repetitive aspect of liturgical prayer that can become rote and hackneyed if one is not careful, it can also be deeply connecting and resonant as well. For some, repeating the prayers that have been prayed for centuries is rooting and comforting. It is a way to join with the multitude of saints — past, present, and future. It can also be promoting unity within the Church, adding an "amen" of agreement to a prayer that resonates with your own current experiences.

Meditative prayer may seem like escape to some and engagement to others. Indeed, anything has the potential to be turned into avoidance. We suggest that meditative prayer can be richly engaging, especially when seen from the vantage point of mindful attention to the present moment. Mindfulness practices are well-documented in the psychological literature as an effective coping strategy and promoter of well-being. Even though mindfulness and meditation are not always synonymous, there can often be overlap. A main thrust behind the robustness of mindfulness is its emphasis on acceptance of the self and the present moment. This acceptance coupled with a meditative attention to core truths and values can be a positive and potent combination. Applied to meditative prayer and experiential avoidance, the practice of simply basking in the presence of God and allowing space to listen and reflect can help to loosen the grip of anxiety and control. Although meditative prayer may not be engaging in a flashy or obvious way, it can be a powerful way of engaging with a difficult situation while releasing the final say to God.

Petitionary prayer seems to have acquired a poor reputation over the several studies that have been referenced, and the lack of significant results in the present study does not help the case. Nevertheless, we are hesitant to label petitionary prayer as immature or deficient. The present study, for example, used items that measure petitionary prayer reference supplication for *material* things for self ("How often do you ask God for material things you may need?") or others ("Ask for material things your friends or relatives may need?"). Although different from Poloma and Pendleton's (1989) original definition, one

wonders whether petitionary prayer could encompass other non-materialistic outcomes (e.g., increased faith, hope, and love) and whether this would demonstrate different relationships with experiential avoidance and mental health outcomes. Furthermore, because Scripture encourages prayer, it is possible that the value of petitionary prayer could be hidden among other virtues that have not been highlighted in these studies. For example, petitionary prayer may be more an exercise in faith and faithfulness rather than a measure of avoidance versus engagement. Indeed, previous studies have found feeling close to God and feeling like they have control through God (God-mediated control) appear to be function as mediators between prayer supplication and well-being, with the former having stronger explanatory ability compared to the latter (Jeppsen et al. 2015). Conversely, another possibility is that petitionary prayer practiced selfishly or inconsiderately has the tendency to draw out pathology in ways that the other prayer types avoid. If a person asks for deliverance or a change in their circumstances but is unwilling or unable to be introspective about their role to play, that petition could become a form of experiential avoidance. Furthermore, although not specific to petitionary prayer, the effects of prayer frequency on psychiatric symptoms/disorders appears to be dependent on one's image about God (e.g., benevolent versus vindictive God; Bradshaw et al. 2008) and attachment style to God (e.g., avoidance versus secure attachment; Ellison et al. 2014). Future research could shed further light into these possible mediation and moderation effects of petitionary prayer on mental health outcomes.

We hope that the results of this study will encourage and support students, religious professionals, and training institutions to continue to do the difficult work of spiritual, character, and moral formation (Porter et al. 2019). As helpful as it is to have longitudinal data to shed light on the process dynamics of spiritual growth and development, one does not need a research study to know that character is formed over the course of a lifetime. To that end, we implore spiritual and religious leaders to reflect deeply on not just how to teach spiritual practices, but also how those spiritual practices form our thoughts, perceptions, and behaviors. We also hope that this research encourages individualized attention in mentorship and discipleship. To try passing on the findings of this study by painting with broad brushstrokes is to risk falling into the trap of shaping behavior without forming character. As has been alluded to several times and demonstrated widely, formation occurs in the context of relationships (Shults and Sandage 2006). A final exhortation is to not lose heart if prayer practices include discomfort. Psychological flexibility comes with growing pains, and engagement is often much more disruptive and unsettling than avoidance. Although it is easy to be preoccupied by the immediate struggle, the benefit comes in the form of long-term mental health.

*Limitations and Directions for Future Research*

Prayer is inherently a difficult construct to study. It is often a very private activity, shaped by culture and tradition, and can be difficult to operationalize for the sake of study. While we feel confident in the items chosen for our prayer type measures for the purposes of this study, future studies may benefit from more extensive measures to draw out further nuances of the multi-dimensions of prayer. For example, the effects of group prayer versus control on mental health appears promising, with experimental evidence suggesting positive effects of communal prayer at post- (Boelens et al. 2009) and 1-year follow-up (Boelens et al. 2012) time points. Given the accumulating evidence for benefits attained through social support (Ellison et al. 2017; Bartkowski et al. 2017), differentiating the effects of private versus communal prayer could be a rich area of study. Participants may also have been tempted to answer items in a socially desirable manner so as not to appear spiritually immature. There is also the possibility of testing fatigue given the comprehensive nature of the survey containing over 100 items and taking approximately 90 minutes to complete.

It is also important to acknowledge that while data was collected from 17 seminaries, there is an immense amount of diversity within Christianity that could not be fully captured

in this study. While we attempted to use inclusive, ecumenical language in this study to increase the potential for utility across denominational lines, we recognize that we are contextualized within our own biases and relational circles. Future studies may consider replicating or extending the findings of this study within different traditions or with an even more comprehensive sample. Moreover, some studies have also compared models across other religions which may help increase understanding of cultural differences (Jeppsen et al. 2022). Seminary students continue to be an understudied population, and more research is needed to further understand the unique challenges they face and how to best prepare them for religious professions.

We also recommend future studies to incorporate positive psychology into their investigation of how to help seminary students thrive. As Proeschold-Bell and her colleagues (2015) have highlighted, mental health and mental illness are correlated, yet separate, continuums, and the integration of both will help provide a more complete understanding of human flourishing. Recognizing this distinction provides some advantages. It enhances precision by revealing significant relationships that might exist on one construct but not the other, thereby being easily missed if not considered, and it helps to provide a more complete picture of a model that benefits from the explanatory variance of both sides (Proeschold-Bell et al. 2015). While we focused on experiential avoidance, anxiety, depression, and work burnout, we recognize that health is not just the absence of ailments but also the presence of strengths, resilience, and virtue.

**Author Contributions:** G.B.L., D.C.W. and E.G.C.; writing—review and editing, G.B.L., D.C.W. and E.G.C. All authors have read and agreed to the published version of the manuscript.

**Funding:** This study received generous funding from the John Templeton Foundation (#61515).

**Institutional Review Board Statement:** This study was approved by the IRB of Biola University (Protocol #SS19-014_SE).

**Informed Consent Statement:** Informed consent was obtained from all subjects involved in the study.

**Conflicts of Interest:** The authors declare no potential conflict of interest with respect to the research, authorship, and/or publication of this article.

## Note

[1] Psychological inflexibility can occur when private events are positive emotions (e.g., avoid feeling elated for fear of disappointment) or when neutral or pleasant internal events decrease people's sensitivity to value-related goals (e.g., fantasizing about a promotion at work which decreases their ability to respond to important responsibilities; Bond et al. 2011).

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
