# Peer review of "Experiential Avoidance Mediates the Relationship between Prayer Type and Mental Health before and through the COVID-19 Pandemic"

_religions, doi:10.3390/rel13070652_

Round 1
Reviewer 1 Report
This article reports on a quantitative study on different types of prayer in relation to mental health among seminary students in de US. The methods are decent, and the study is conducted well; the study results are reported on systematically. The paper is publishable and offers interesting insight into new research on the role of prayer within Christan seminary. I think some minor points might be improved.
1) The introduction is interesting and opens the paper, though it is a bit long. I think a critical look at the text and length would improve the author(s) argument in the introduction. What is the main point the author(s) want to make? Try to interrelate the different paragraphs and concepts from the introduction more. Some sources are bit outdated, try to use more recent literature in the introduction to position the paper in current debates.
2) Also conceptualize the different types of prayer shortly in the introduction.
3) Define “mental health” explicitly in the introduction (also how it relates to experiential avoidance).
4) This comment is optional, but I think would make the argument in the introduction stronger: how is the stress level related to other “professions”? E.g. nurses or teachers? Why are clergy a specific group? The authors do tell us something about that but could make that a bit more explicit.
5) The methods and results seem decent from what I can tell.
6) Try to engage more with the conceptual literature in the discussion. Make more use of references to give the study a broader outline at the end.
I have some minor points:
-Table 1 is missing a title
-There has been a lot of research conducted on prayer that is not quoted here: add some more sources to conceptualize prayer more in the introduction, here are some suggestions:
Illueca, Marta, and Benjamin R. Doolittle. 2020. “The Use of Prayer in the Management of Pain: A Systematic Review.” Journal of Religion and Health 59 (2): 681–99. https://doi.org/10.1007/s10943-019-00967-8.
Masters, Kevin S., Ralph W. Emerson, and Stephanie A. Hooker. 2020. “Effects of Devotional Prayer and Secular Meditation on Cardiovascular Response to a Faith Challenge Among Christians.” Psychology of Religion and Spirituality. https://doi.org/10.1037/rel0000369.
Pandya, Samta P. 2018. “Prayer Lessons to Promote Happiness Among Kindergarten School Children: A Cross-Country Experimental Study.” Religious Education 113 (2): 216–30. https://doi.org/10.1080/00344087.2017.1393181.
Plante, Thomas G. 2022. “Using the Examen, a Jesuit Prayer, in Spiritually Integrated and Secular Psychotherapy.” Pastoral Psychology 71 (1): 119–25. https://doi.org/10.1007/s11089-021-00967-0.
Silton, Nava R., Kevin J. Flannelly, Kathleen Galek, and David Fleenor. 2013. “Pray Tell: The Who, What, Why, and How of Prayer Across Multiple Faiths.” Pastoral Psychology 62 (1): 41–52. https://doi.org/10.1007/s11089-012-0481-9.
Reviewer 2 Report
This paper examines how experiential avoidance mediates the practice of various types of prayer and mental health outcomes among seminarians. The sample is impressive and the study provides insight into an important issue. The paper will be enhanced by addressing the following considerations. They are mentioned in no particular order.
1. Some conceptual or theoretical grounding in the stress process model might be useful here to shed additional explanatory light on the findings. See, e.g., https://www.mdpi.com/2077-1444/8/9/195/htm. There is also research on positive and negative forms of religious coping. I am not suggesting a review of this empirical work, as it is extensive. Rather, the use of these concepts (as theory) may in the interpretation of results. Findings are most compellingly interpreted using theory.
2. The body of the paper really does not specify the research problem or clearly convey the significance of the study early on in the manuscript. These are vital. Solid articles feature an introduction that provides a bit of general context before turning to the research problem and its significance. Only then is the literature reviewed. This paper dives directly into the literature review and then states the research problem on pages 6-7. That seems delayed. See either of the papers recommended in this review for a more traditional structure.
3. Experiential avoidance is defined in a potentially contradictory fashion. "Experiential avoidance is the attempt to escape from or control difficult, uncomfortable, or undesirable situations..." Escaping and controlling are two very different reactions. This contradiction needs to be resolved. If the problem is with the original conceptualization, highlight and resolve this problem in your text, noting your departure from the original formulation in a footnote.
4. On this same point, is the concept of experiential engagement the obverse of experiential avoidance? If so, experiential engagement might be introduced earlier because this study is really examining the absence of experiential avoidance. Thus, it may be examining experiential engagement. I realize the authors must stick to the scale they've used and avoidance is commonly studied. So, that terminology is needed.
5. What other variables, if any, were collected? If they were collected, should they not be included as controls? I am thinking about denomination of the seminarians, a positive personal test for COVID, or loss of loved one due to COVID. Sociodemographic variations, at the very least, should be considered so hypotheses can be tested net of controls. The models seem to be focused on zero-order relationships without attention to potential confounding factors.
6. The study would be more compelling if it first examined how the overall volume (frequency) of any prayer (all types combined) first affected the relationship of interest, after which individual prayer types could be examined. (Perhaps the analysis I'm seeking could be put in an appendix.) I am not asking for an overhaul on this point. But methodologically, it would demonstrate that quantity alone is not what matters, but rather type of prayer (a form of quality). Relatedly, there is evidence that communal prayer has different effects than individual prayer in relation to mental health (https://www.mdpi.com/2077-1444/8/9/191). Can this be explored or added to the future research? Seminarians likely engage in lots of communal prayer.
7. Should not more emphasis be placed on the time series data? This facet of this rich data set seems underutilized.
There are some small typos here and there (e.g., bottom of page 8, p<.01 should be p < .01 as is done with the preceding line where p < .05). Generally sound work with some room for improvement.
Round 2
Reviewer 2 Report
I commend the authors on a thorough revision. I have no further comments.